# Optimization Strategy of Rolling Mill Hydraulic Roll Gap Control System Based on Improved Particle Swarm PID Algorithm

**DOI:** 10.3390/biomimetics8020143

**Published:** 2023-03-31

**Authors:** Ying Yu, Ruifeng Zeng, Yuezhao Xue, Xiaoguo Zhao

**Affiliations:** 1School of Mechanical and Electrical Engineering, Xi’an University of Architecture and Technology, Xi’an 710055, China; 2Capital Engineering & Research Incorporation Limited, Beijing 100176, China

**Keywords:** PLC, medium and thick plates, roll gap control, improved particle swarm algorithm, semi-physical simulation

## Abstract

Medium and heavy plates are important strategic materials, which are widely used in many fields, such as large ships, weapons and armor, large bridges, and super high-rise buildings. However, the traditional control technology cannot meet the high-precision control requirements of the roll gap of the thick plate mill, resulting in errors in the thickness of the medium and heavy plate, thereby reducing the quality of the product. In response to this problem, this paper takes the 5500 mm thick plate production line as the research background, and establishes the model of the rolling mill plate thickness automatic control system, using the Ziegler–Nichol response curve method (Z-N), particle swarm optimization (PSO) algorithm and linear weight particle swarm optimization (LWPSO) algorithm, respectively, optimizes the parameter setting of the PID controller of the system, and uses OPC UA communication technology to realize the online semi-physical simulation of Siemens S7-1500 series PLC (Siemens, Munich, Germany) and MATLAB R2018b (The MathWorks, Natick, Massachusetts, United States). Comparative studies show that when the same roll gap displacement step signal is given, the overshoot of the system response using the LWPSO algorithm is reduced by 14.26% and 10.18% compared with the Z-N algorithm and the PSO algorithm, and the peak time is advanced by 0.31 s and 0.05 s. The stabilization time is reduced by 3.71 s and 4.31 s, which effectively improves the control accuracy and speed of the system and has stronger anti-interference ability. It has certain engineering reference and application value.

## 1. Introduction

Heavy industry has always played an extremely important role in the continuous development of the country. With the rapid progress of science, technology and economy, higher requirements have been put forward for the quality of industrial products. For example, aerospace materials must be able to withstand high stress and inertial forces, as well as high-speed rotation of rotating parts; laying natural gas, oil and other energy transmission pipelines requires a large number of high-quality and reliable pipeline steel plate; the deck of the ship must be able to withstand the huge impact of frequent landings of carrier-based aircrafts; the steel plates used for tanks and armored vehicles must be able to withstand the penetration of armor-piercing bullets; the steel plates used in the construction of large bridges must be able to withstand the frequent vibrations of trains running at high speeds, etc. These scenarios require a large number of high-quality medium and thick plates of different specifications. Among the quality indicators of medium and thick plates, thickness is one of the most important quality indicators [1]. Small thickness deviations may cause serious consequences in the above scenarios. The thickness accuracy of medium and thick plates is related to the sustainable development of related industries [2].

The thicknesses of the plate and strip mainly depend on the height of the roll gap between the two working rolls of the rolling mill. The essence of controlling the thicknesses of the plate and strip is to control the height of the roll gap. Therefore, the rolling mill hydraulic automatic thickness control system plays a very important role in the steel rolling process. In the process of continuous technological advancement, there are also higher requirements for the accuracy of thickness control [3,4]. However, due to internal and external factors such as nonlinearity, strong coupling, and changes in temperature and humidity around the equipment in the actual rolling mill system, it is difficult to obtain accurate system control parameters by using the traditional PID control method in engineering, resulting in the hydraulic servo screwdown device not being able to achieve the specified control accuracy when controlling the displacement of the working roll, so there is an error in the height of the roll gap, and the thickness quality of the produced plate cannot meet the process requirements [5,6]. In response to such problems [7,8], many modern control strategies have emerged in recent years; that is, various intelligent algorithms are introduced into the PID control to improve the control accuracy and practicability of the system [9,10].

In [11], the single neuron was combined with the PID control and they were applied in the rolling mill hydraulic automatic thickness control system. The results showed that compared to the traditional PID control, the control accuracy and anti-interference ability were improved. In [12], the improved Smith fuzzy PID algorithm was used to tune the PID parameters. Compared to the conventional PID algorithm, the overshoot was reduced, and the stability and self-adaptive ability were improved. In [13], the ANSYS software for finite element simulation is used, using full simulation to study the product quality problems caused by large residual stress in the heat treatment process of seamless steel pipe rolling. Using this approach can shorten research time and reduce research costs. Reference [14] proposes a hybrid intelligent PID controller to improve the performance of the steel rolling process; this new hybrid system combines rule-based systems and neural networks, using long-term field data sets combined with artificial neural networks to optimize the parameters of the PID controller. Compared with the traditional PID controller, the control accuracy and speed have been greatly improved.

Based on the reviewed literature, the above methods still have problems such as insufficient convergence speed, too many iterations, and are easy to fall into local optimal solutions when optimizing the PID parameters of the control system, and digital simulation is used for verification in the simulation test. Digital simulation has high requirements on the accuracy of modeling, and the accuracy of the model is closely related to the credibility of the conclusion. However, the rolling mill hydraulic roll gap control system studied in this paper is a complex nonlinear system, for which is difficult to obtain accurate model. Aiming at the above problems, combined with the actual production situation of the project site, a mathematical model of the hydraulic roll gap control system of the 5500 mm rolling mill is established. The linear weight particle swarm optimization (LMPSO) algorithm is used to complete the tuning of the system PID parameters, and the weight is linearly modified during the optimization process to achieve rapid convergence, accurately locating the position of the global optimal solution. Additionally, the use of the OPC UA communication technology to realize the semi-physical simulation of Siemens S7-1500 series PLC and MATLAB/Simulink, combine the advantages of digital simulation and physical simulation, shorten the research cycle, reduce research costs, and improve the reliability of simulation, determines the applicability of this optimization algorithm to the rolling mill hydraulic roll gap control system, which can effectively solve the problem of roll gap control accuracy and improve the quality of thick plate production.

The rest of the paper is organized as follows: Section 2 introduces the process flow of thick plate rolling, the structure of thick plate rolling mill and rolling principle. The modeling of the rolling mill hydraulic roll gap control system is given in Section 3. Details of particle swarm optimization and the proposed linear weight particle swarm optimization algorithm are given in Section 4. Section 5 introduces the principle and construction of a semi-physical simulation platform. Section 6 presents the results and discussion. Finally, Section 7 summarizes the work and suggests future directions.

## 2. Process and Principle of Heavy Plate Rolling

Most of the early rolling mills used the electric roll gap control system (EGC), which mainly completed the pressing operation by a large motor, a reducer, and a pressing screw. The equipment structure was complex and the precision was low. It was gradually replaced by the hydraulic roll gap control (HGC) system with superior control performance, forming the automatic thickness control system (AGC) of the rolling mill [15].

The general process flow of medium and heavy plate rolling is shown in Figure 1. The raw materials are firstly dephosphorized by the heating furnace and high-pressure water, and then sent to the rolling mill by the roller table for rough rolling and finishing rolling. After controlled cooling, the rolled steel billet is sent to the straightening machine to control and correct the shape of the plate, then to cut the head, double sides, and cut to a length that meets the specifications required by the process [16]. Finally, heat treatment is used to increase the strength of the strip steel, improve its toughness, and eliminate residual stress; the main process has been completed.

The hydraulic rolling mill in the process flow is the core equipment in the rolling process, which has a direct impact on the thickness of the plate and strip. There are two installation methods of HGC hydraulic cylinders in the thick plate rolling mill frame, the upper type and the lower type [17]. The 5500 mm thick plate rolling mill studied in this paper adopts the bottom installation, as shown in Figure 2 and Figure 3, the HGC hydraulic cylinder is installed on the bottom of the frame, and above it is the lower support roll, the lower work roll, the upper work roll, and the upper work roll. The roll gap of a rolling mill generally refers to the height between the upper and lower work rolls.

During the rolling process, the rolled piece is sent from the roller table to the upper and lower work rolls of the rolling mill, is compressed by the rolling force, and undergoes plastic deformation. Under the interaction of forces, the body of the rolling mill will also be affected by the same force, so that the actual roll gap is larger than the unloaded roll gap during the rolling process, resulting in fluctuations in the thickness of the rolled piece. This phenomenon is called the bouncing phenomenon [18,19]. As shown in Figure 4, the original thickness of the rolled piece is H, the initial roll gap height is $\;h_{0}$, the actual roll gap height is h.

According to the principle of mechanics, the algebraic relationship between variables can be written as the bouncing equation:(1)$$h=h_{0}+\frac{P}{K}$$
where

P is the rolling force.

K is the stiffness of the rolling mill.

In order to avoid this problem, generally at the entrance of the rolling mill, the thick ness fluctuation of the incoming material is predicted according to the process, and the pressure is adjusted to eliminate the thickness difference caused by the bouncing of the rolling mill to ensure that the plate thickness meets the production accuracy requirements.

Therefore, the adjustment of the height of the roll gap is also a high-precision control of the reduction system combined with the rolling mill bounce equation, which is an indispensable research topic in the rolling production of medium and heavy plates.

## 3. Modeling of Rolling Mill Hydraulic Roll Gap Control System

The hydraulic AGC system adjusts the oil flow to drive the hydraulic cylinder by controlling the opening of the electro-hydraulic servo valve, and adjusts the height of the roll gap through the vertical displacement of the hydraulic cylinder, so as to realize the precision control of the thicknesses of the plate and strip.

The system consists of multiple control loops. The high-precision control of the thicknesses of the plate and strip is completed through the position closed-loop, rolling pressure closed-loop, and thickness gauge closed-loop, thereby eliminating the deviation in thickness. Among them, the full name of the HGC system is the hydraulic roll gap automatic control system. It satisfies the real-time dynamic adjustment of the small stroke roll gap through the position closed-loop, and has the technological requirements of high precision and fast response. It is the core component of the AGC system of the rolling mill.

According to the control block diagram in Figure 5, it can be seen that the system is divided into five main parts, and then, mathematical modeling is carried out on them.

### 3.1. Servo Amplifier

The servo amplifier is a kind of amplification conversion element. The deviation signal can be amplified and converted into the required hydraulic signal. In the electro-hydraulic servo valve, the amplifier generally adopts deep negative feedback [20], and its response speed is fast and the frequency is high, so it can be regarded as a proportional link. The amplification gain is represented by $K_{a}$, The transfer function of is as follows:(2)$$G_{1}\left(s\right)=K_{a}$$

### 3.2. Electrohydraulic Servo Valve

The electro-hydraulic servo valve is usually composed of an electrical conversion device and a hydraulic amplifying device [4]; its main function is to connect the electrical system with the hydraulic system; it can carry out continuous two-way control on pressure or flow at the same time, and has the advantages of fast response speed and high control precision, and is the core of the hydraulic control system.

The electro-hydraulic servo valve converts the tiny input current signal to a high-power hydraulic output signal, such as pressure and flow. According to the working principle of the electro-hydraulic servo valve, ignoring the small effects of oil leakage, mechanical wear, pressure consumption, etc., it is assumed that the structure of the valve is symmetrical, the flow coefficient is the same as the zero-opening slide valve, and the oil supply pressure $P_{s}$ is a constant value. When the spool generates zero displacement $x_{v}$, the calculation formula of the oil flow $\;Q_{1}\;$ flowing into the valve and the oil flow $Q_{2}$ flowing out of the valve are as follows [21]:(3)$$Q_{1}=C_{d}Wx_{v}\sqrt{\frac{2}{\rho }\left(P_{s}-P_{L}\right)}$$
(4)$$Q_{2}=C_{d}Wx_{v}\sqrt{\frac{2}{\rho }P_{L}}$$
where

$P_{s}$ is the oil supply pressure.

W is the gradient area of the valve port.

$C_{d}$ is the flow coefficient.

$x_{v}$ is the displacement of the servo valve spool.

ρ is the fluid density.

$P_{L}$ is the load pressure.

It can be seen in Formula (4) that the flow–pressure characteristic equation of the electro-hydraulic servo valve is a nonlinear composite function. In the process of modeling, it is necessary to avoid nonlinear terms as much as possible. Combined with engineering practice, when the rolling mill is working normally, the working area of the electro-hydraulic servo valve is usually near zero, and the change is not large, so it can be approximately linearized.

In the rolling mill hydraulic AGC system, the transfer function of the servo valve is usually a second-order oscillation link [22], namely
(5)$$G_{2}=\frac{Q\left(s\right)}{I\left(s\right)}=\frac{K_{v}}{\frac{s^{2}}{\omega_{v}^{2}}+\frac{2\xi_{v}}{\omega_{v}}s+1}$$
where

$K_{v}$ is the flow gain coefficient of the servo valve.

Q is the output flow of the servo valve.

I is the input current of the servo valve.

s is the Laplacian operator.

$w_{v}$ is the natural frequency of the servo valve.

$\xi_{v}$ is the damping coefficient.

### 3.3. Valve-Controlled Hydraulic Cylinder

The hydraulic system is the power core of the entire AGC system, and the servo-valve-controlled hydraulic cylinder is the most direct source of power and plays a central role in the system. Similar to the servo valve, its working area is concentrated near the zero point, and the simplified linear flow equation can be obtained by using the Taylor series expansion
(6)$$Q_{L}=K_{q}X_{v}-K_{c}P_{L}$$

According to Formula (6), the flow equation of the hydraulic cylinder can be written as
(7)$$Q_{L}=A_{p}x_{p}s+CP_{L}+\frac{V_{0}}{\beta_{e}}P_{L}s$$
where

$K_{q}$ is the flow amplification coefficient.

$K_{c}$ is the flow pressure amplification coefficient.

$A_{p}$ is the effective area of the piston.

$x_{p}$ is the effective displacement of the piston.

C is the total leakage coefficient of the hydraulic cylinder.

$\beta_{e}$ is the elastic modulus of the hydraulic oil.

$V_{0}$ is the total volume of the hydraulic cylinder.

### 3.4. Mill Roll System

The hydraulic components will be affected by the load characteristics when performing production work, and the rolling mill is a very complex dynamic system, so it needs to be simplified into a single degree of freedom system during calculation, when various loads are equivalent to the hydraulic piston, the force received by the piston can be called the external load force. According to Newton’s law, the balance equation between the system and the external load force can be written as
(8)$$P_{L}=\frac{1}{A_{p}}(m_{t}s^{2}x_{p}+Kx_{p}+F)+\frac{A_{b}}{A_{p}}P_{b}$$
where

$A_{b}$ is the piston area of the rod cavity.

$P_{b}$ is the piston pressure of the rod cavity.

$m_{t}$ is the total weight of the load and the piston.

$B_{p}$ is the total damping coefficient of the load and the piston.

K is the load elastic coefficient.

F is the external load force.

### 3.5. Displacement Sensor

The displacement sensor is used in the hydraulic AGC system to measure the displacement of the hydraulic cylinder piston during the working process, which is one of the most important feedback links of the plate thickness control system, which is usually regarded as a first-order inertial or proportional element, namely
(9)$$G_{w}(s)=\frac{U_{f}}{X_{p}}=\frac{K_{w}}{1+T_{w}s}$$
where

$K_{w}$ is the sensor gain coefficient.

$T_{w}$ is the sensor time constant.

$U_{f}$ is the output voltage of the sensor.

According to the Formulas (5)–(9) and combined with the parameters of each main component of the 5500 mm plate rolling mill in Table 1, by calculating the parameters in the model and drawing the hydraulic AGC system block diagram, we arrive at the system control block that is seen in Figure 6.

## 4. Linear Weight Particle Swarm Optimization (LWPSO)

The particle swarm optimization (PSO) algorithm was proposed by Ebarhart RC and Kennedy J in the 1990s, inspired by the flocking activities of flying birds in nature [23,24]. PSO simulates the group behavior of birds when they are foraging. The algorithm is easy to implement, has high precision and fast convergence speed, and has been widely used in solving practical engineering problems.

The basic idea of the PSO algorithm is similar to the decision-making process of birds foraging. In the process of foraging birds, each bird has a strong randomness, and its flight speed and direction are random, so in the initial stage, the whole flock of birds is distributed disorderly in the foraging space, but as time goes by, each bird will transfer location information to other birds and they will learn from each other. Then, the next action is an “experienced” behavior. When a bird finds the food position, it finds a local optimal solution, and the position all birds find is the global optimal solution. The global optimal solution is the trend of the entire flock of birds moving, and it is constantly approaching the global optimal solution as time runs out and the number of iterations increases [25].

The state of the particle is analyzed through the particle’s position, speed, and fitness parameters, and the fitness value corresponding to each particle is used to judge whether the particle is closest to the global optimum [26,27].

Each bird in the flock is regarded as a particle, and the flock of birds is a particle swarm, and the particle swarm with M particles is optimized in the *N*-dimensional space; then, the position of particle i in the *N*-dimensional space is expressed as [28]
(10)$$x=\left(x_{i1},x_{i2},x_{i3},\ldots ,x_{iN}\right)$$

The optimal position that the particle has reached is indicated as
(11)$$p_{i}=\left(p_{i1},p_{i2},p_{i3},\ldots ,p_{iN}\right)$$

The velocity of the particle when it reaches the optimal position is expressed as
(12)$$v_{i}=\left(v_{i1},v_{i2},v_{i3},\ldots ,v_{iN}\right)$$

Finally, the optimal position passed by each particle is expressed as
(13)$$p_{g}=\left(p_{g1},p_{g2},p_{g3},\ldots ,p_{gN}\right)$$

The update method of particle position and velocity is indicated as
(14)$$\begin{array}{l} v_{id}=wv_{id}+c_{1}r_{1}\left(p_{id}-x_{id}\right)+c_{2}r_{2}\left(p_{gd}-x_{gd}\right)\\ x_{id}=x_{id}+v_{id} \end{array}$$
where w is the weight factor, $c_{1}$ and $c_{2}$ are learning factors, $r_{1}$ and $r_{2}$ are random numbers between [0, 1].

The main implementation steps of the PSO algorithm are shown in Figure 7.

Due to the characteristics of the PSO algorithm itself, it is easy to fall into the predicament of local optimum when searching for optimization; aiming to address this problem, an improved PSO algorithm is proposed, which can continuously adjust the weight of the algorithm during the optimization process. When the weight is large, it can quickly perform global optimization in the early search to lock the optimal range. When the weight is small, it can perform targeted local optimization in the later search to lock the global optimum, avoiding the phenomenon of premature algorithm. As the number of iterations increases, the weight will gradually decrease until the global optimum is obtained, and the relationship is as follows [28]
(15)$$w_{0}=w_{2}+\left(w_{1}-w_{2}\right)\left(1-\frac{K}{T}\right)$$
where

$w_{0}$ is the actual weight.

$w_{1}$ is the starting weight.

$w_{2}$ is the ending weight.

K is the current number of iterations.

T is the total number of iterations.

In this paper, when the linear weight particle swarm optimization algorithm is used to optimize the control parameters, the initial weight is $\;w_{1}$ = 0.9, the end weight is $w_{2}$ = 0.4, and total iterations are *T* = 100.

As shown in Figure 8, the optimization of the PID parameters by the improved particle swarm optimization algorithm is to input the system error as the fitness of each particle in the optimization algorithm and calculate its value; then, continuously adjust the PID parameters according to the obtained fitness value in order to achieve the goal of optimal control of the system.

## 5. Construction of Semi-Physical Simulation Platform

According to the established transfer function of each component, the closed-loop transfer function of the rolling mill roll gap control system can be written as
(16)$$G\left(s\right)=\frac{0.855s+85.48}{4.87\times 10^{-11}s^{6}+3.88\times 10^{-8}s^{5}+2.41\times 10^{-5}s^{4}+0.00575s^{3}+1.37s^{2}+100s+85,480}$$

Using the transfer function to establish the rolling mill hydraulic HGC system model through the MATLAB/Simulink software can be seen in Figure 9.

The plate mill model of the simulation platform is established in the Simulink software, while the physical controller uses a Siemens S7-1500 series large-scale programmable logic controller (PLC). This model is widely used in the field process control of steel rolling, using the Profinet industrial bus standard to connect the hardware channel between the upper computer model and the lower computer PLC.

Figure 10 is a schematic diagram of the semi-physical simulation platform. The rolling mill model is established in the upper computer Matlab/Simulink software, and the lower computer S7-1500 series PLC realizes the control of the rolling mill model through the TIA Portal software. The ET 200SP controller and ET 200MP module of the distributed I/O system are also connected to the system as backups, and are put into use when the experiment requires remote control or the I/O address of the main PLC is insufficient. All experimental equipment is interconnected with switches through industrial Ethernet.

To communicate between software and hardware through OPC UA, firstly an OPC server in the TIA Portal of Siemens PLC engineering configuration software are established, and input variables are created; secondly, an OPC UA Client channel in the KepServer software are established; finally, OPC Read, OPC Write modules and OPC Config data transmission modules are established in the rolling mill model of Matlab/Simulink. In the simulation process, the set roll gap value variable output in the PLC is used as the input of the rolling mill model, and the roll gap deviation signal output by the model is returned to the PLC as an input signal to form a closed feedback loop. the semi-physical simulation platform has been completed.

Figure 11a,b are the physical diagram and wiring diagram of the simulation platform. The layout and design of the platform are inspired by the electrical control cabinet on site. Device 1 in the Figure 11a is an air circuit breaker, which is used for the short-circuit protection and leakage protection of the simulation platform to prevent equipment from being damaged due to short-circuits and to avoid personal safety accidents caused by leakages. Device 2 is the Siemens ET-200sp controller, which is usually used as a distributed I/O system, and is used for distributed control when there are many control devices and the device layout is scattered in large-scale industrial control sites; it can solve the local distribution of I/O and form a ring network with the main PLC to complete more efficient and convenient on-site control; Device 3 is Siemens S7-1500 series PLC and its power supply module; the rectifier transformer in the power supply module can output 24 V direct current to be used by various equipment on the platform; Device 4 are digital and analog input/output modules, which are used for remote data exchange between controlled equipment and PLC; Device 5 is a switch used for communication between the host computer and the lower computer of the simulation platform; Device 6 is a servo drive, used to control the servo motor, and it can cooperate with PLC to carry out some simple control experiments; Device 7 is a relay module for signal switching, monitoring and control; Device 8 is a connecting terminal, which is used to connect the signal lines inside and outside the cabinet, it can also make the electrical wiring more beautiful, and facilitate the maintenance and repair of the staff; Device 9 is a signal jammer, which prevents mutual interference between devices in the cabinet.

## 6. Result and Discussion

The PID controller is widely used in industrial process control, where P (proportion) is the proportional link, I (integral) is the integral link, and D (differential) is the differential link. The tuning of PID parameters has always been a research problem in engineering practice, the early PID parameter tuning mainly used the cut-and-try method, and the tuning was carried out based on the experience of field workers. However, this method has many uncertain factors and has a great relationship with the accumulation of experience of the staff, so it was gradually replaced by the Ziegler–Nichol response curve method. The Z-N tuning method is still widely used today, and it is used in engineering projects as a conventional PID control, the improved particle swarm optimization algorithm used in this paper can further improve the stability and control accuracy on the basis of the conventional Z-N algorithm and particle swarm optimization algorithm control.

In the simulation, the PLC gives the displacement of the upper roller system input by the system as a pulse signal with an amplitude of 0.7 cm and a period of 2 ms based on the on-site process model. It can simulate the control requirements for special plate shapes during production and compare and analyze the system response results under different control algorithms.

First, no optimization method for the system parameters should be used, and the given signal should be input into the original system model because the rolling mill hydraulic roll gap control system has the characteristics of nonlinearity and strong coupling. Its response is shown in Figure 12; the accuracy of the response curve is very low, the followability is poor, and the amplitude error is about 70%. The response speed does not meet the production process requirements and it is difficult to reflect the changes of the pulse input signal, so it is necessary to introduce an algorithm to correct its control performance.

Figure 13 uses the Ziegler–Nichol response curve method to adjust the three parameters of the PID and substitute them into the system; after the simulation operation, it can be seen in the response curve that its control curve tracking is obviously better than the original response of the system, its amplitude error is about 4.28%, which can basically realize the following of the given signal of the roll gap, but its response speed is slow, and there is obvious hysteresis in the wave tail part of the response. There is still a significant difference between the fitting degree and the given input, and it still cannot meet the requirements of the project’s thick plate rolling precision and plate surface technology.

The response curve that is produced when the LMPSO algorithm is used to tune the system parameters, is shown in Figure 14; it can be obtained from the curve that the amplitude error of the response can be stabilized within 1%, the system hysteresis is small, and the response speed also meets the process requirements of the project.

The response curve basically achieves a high degree of fitting. Compared with the Z-N tuning method, the followability is greatly improved, and the control performance is also superior; for the pulse signal, it can reflect the real-time changes of the input signal almost perfectly, meeting the control precision requirements in the rolling process to ensure the quality of the product.

Aiming at the roll gap displacement step signal commonly used in the rolling process model as the system input, the simulation is also carried out; given a step signal with a displacement of 0.1 cm for the upper roll system, three different tuning methods of the PID parameters are used to simulate and compare their response curves.

It can be seen in the simulation response curves in Figure 15, Figure 16 and Figure 17 and Table 2 of the parameters of the three strategies, that in the semi-physical simulation of the rolling mill hydraulic roll gap control system, compared with the Z-N algorithm, the maximum overshoot of the PSO algorithm and the LMPSO algorithm were reduced by 10.18% and 14.26%; compared with the first two control methods, the LMPSO algorithm shortens the system stabilization time by 3.71 s and 4.31 s; the peak times are reduced by 0.31 s and 0.05 s, respectively. The Z-N algorithm is shorter than the PSO algorithm in the stability time, but the overshoot is too large. Only the LMPSO algorithm has the best performance indicators.

In order to verify the anti-interference ability of the system, in the stable state of the system, an interference signal with an amplitude of 0.01 cm is added at 10 s, as shown in Figure 18 and Figure 19. Under conventional PID control, the system’s ability to suppress interference signals is poor, and the system tends to be stable 9 s after being disturbed. Under the control of the LMPSO algorithm, the ability to suppress the interference signal is strong, and the amplitude of the interference signal begins to attenuate when it does not reach 0.01 cm, and the system tends to be stable after being disturbed for about 5 s, which reflects its good anti-interference.

Under the control of the LMPSO algorithm, both the control accuracy and the response speed are improved to varying degrees; it can better adapt to the control precision and speed requirements of modern rolling. The stability of the HGC system of the rolling mill is optimized, which is conducive to the improvement of product quality, and at the same time, ensures the yield of rolling.

## 7. Conclusions

This paper improves the PSO algorithm by linearly changing the weight size. In the initial stage of optimization, a larger inertia weight is set to improve the global search ability and quickly locate the global optimal region. With the advancement of the optimization process, the inertia weight decreases linearly, the local search ability gradually becomes stronger, the convergence speed becomes faster, and the global optimal solution is quickly found in a small range, through the control system model and PLC in the MATLAB/Simulink environment for semi-physical simulation. The results show that compared with the Z-N response curve method and PSO algorithm, the LMPSO algorithm used in this paper can respond to the given roll gap displacement of the system faster and more accurately, reduce the roll gap error, improve the accuracy of the rolling mill roll gap control, and better adapt to the continuous change of roll gap height in the continuous rolling process of medium and heavy plates, so as to improve the thickness quality of plates. At the same time, combined with the advantages of semi-physical simulation, the test can be carried out in the environment that meets the overall performance index of the system, which improves the reliability of the test results and has good engineering application value.

## Figures and Tables

**Figure 1 biomimetics-08-00143-f001:**
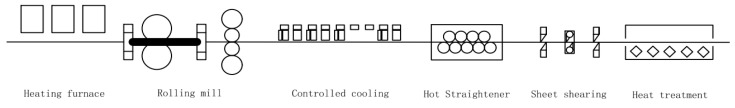
Process flow chart of thick plate rolling.

**Figure 2 biomimetics-08-00143-f002:**
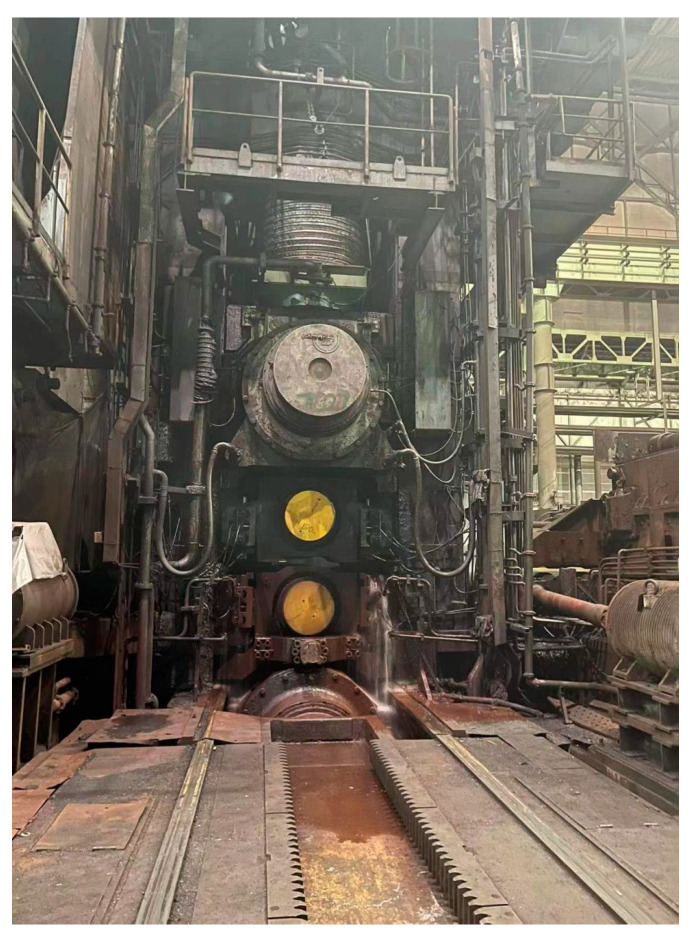
A 5500 mm thick plate rolling mill physical map.

**Figure 3 biomimetics-08-00143-f003:**
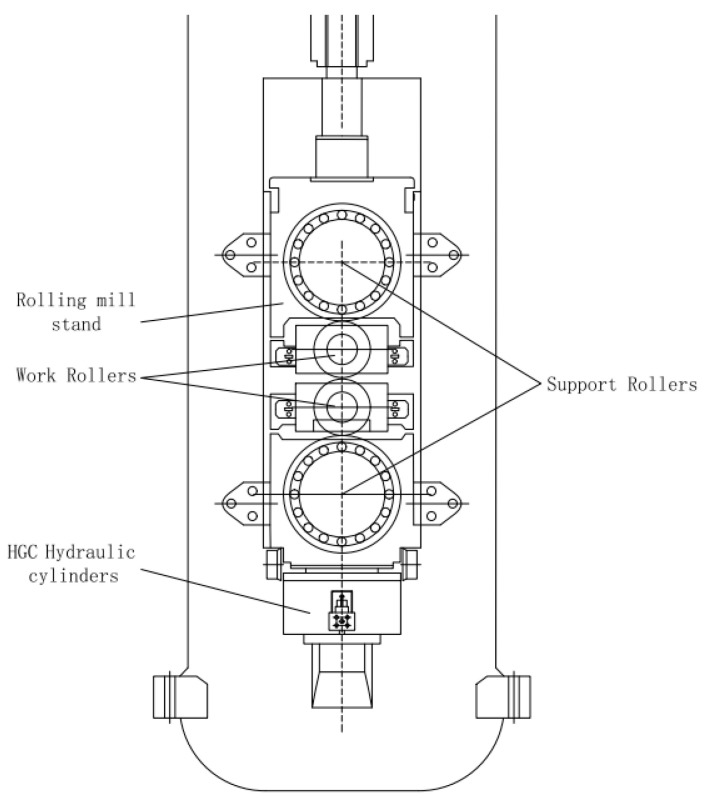
Structural schematic diagram of 5500 mm thick plate rolling mill.

**Figure 4 biomimetics-08-00143-f004:**
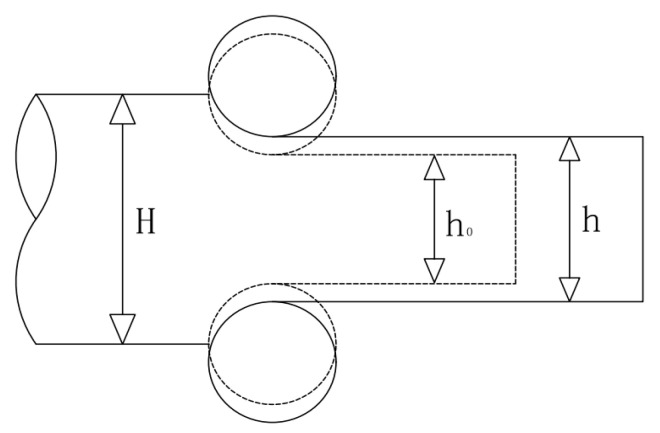
Bounce phenomenon.

**Figure 5 biomimetics-08-00143-f005:**
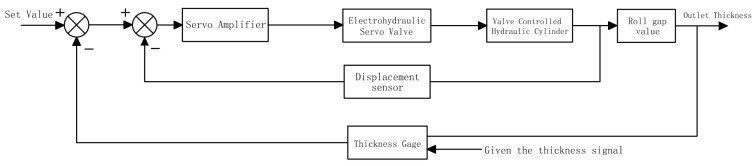
Block diagram of system control.

**Figure 6 biomimetics-08-00143-f006:**
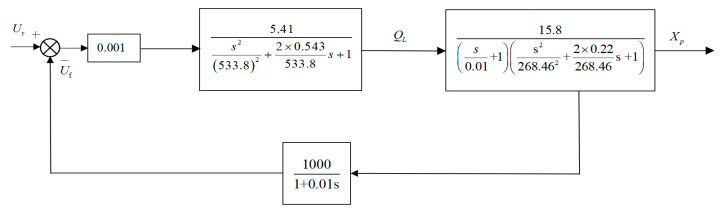
System control block diagram.

**Figure 7 biomimetics-08-00143-f007:**
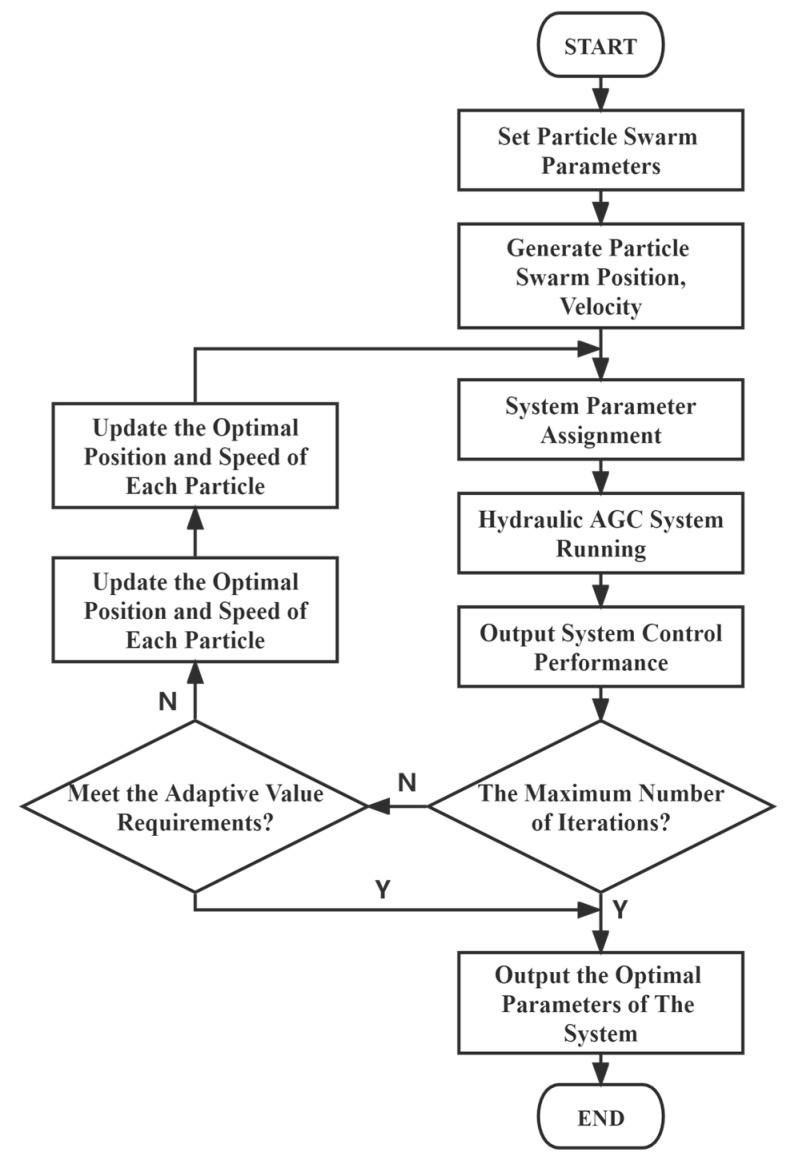
Algorithm flowchart.

**Figure 8 biomimetics-08-00143-f008:**
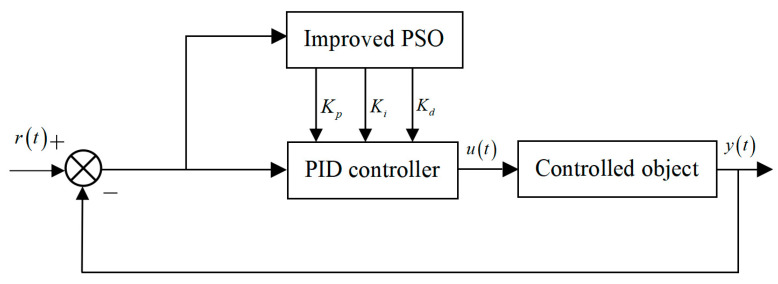
Schematic diagram of PID parameter tuning of improved particle swarm algorithm.

**Figure 9 biomimetics-08-00143-f009:**
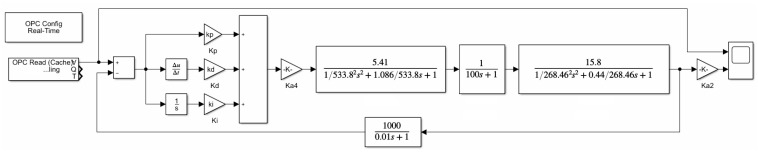
Rolling mill hydraulic HGC system modeling.

**Figure 10 biomimetics-08-00143-f010:**
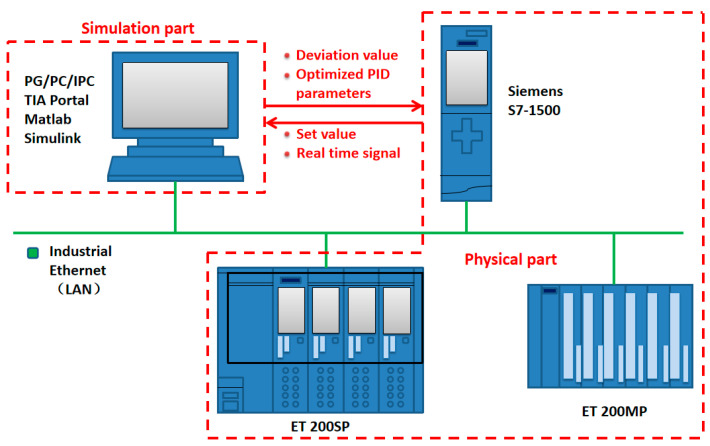
Schematic diagram of the simulation platform.

**Figure 11 biomimetics-08-00143-f011:**
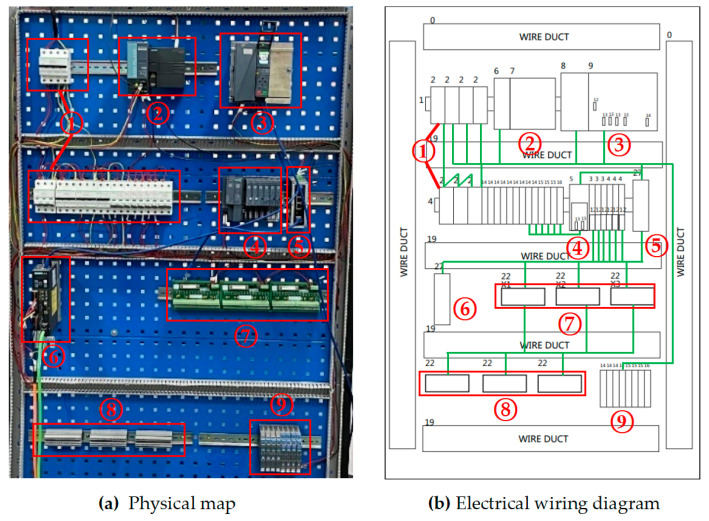
Simulation platform physical diagram and wiring diagram.

**Figure 12 biomimetics-08-00143-f012:**
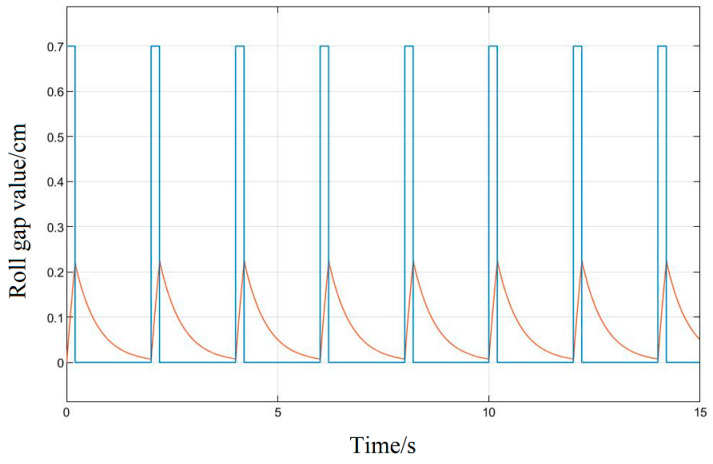
System raw impulse response diagram.

**Figure 13 biomimetics-08-00143-f013:**
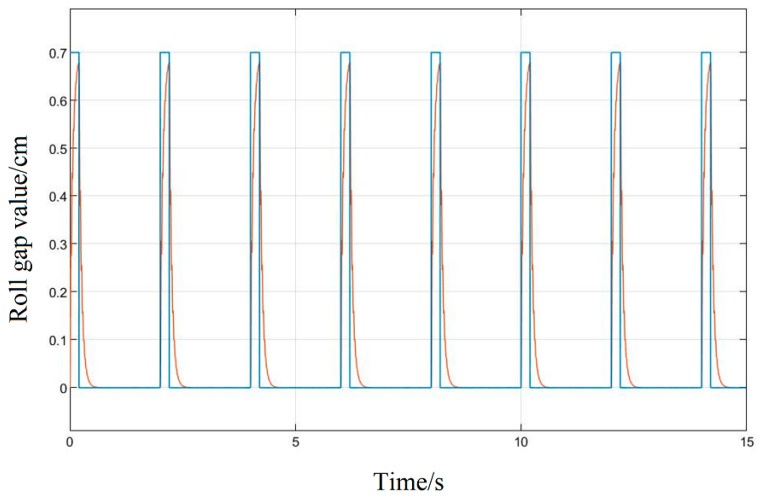
Z-N method system impulse response diagram.

**Figure 14 biomimetics-08-00143-f014:**
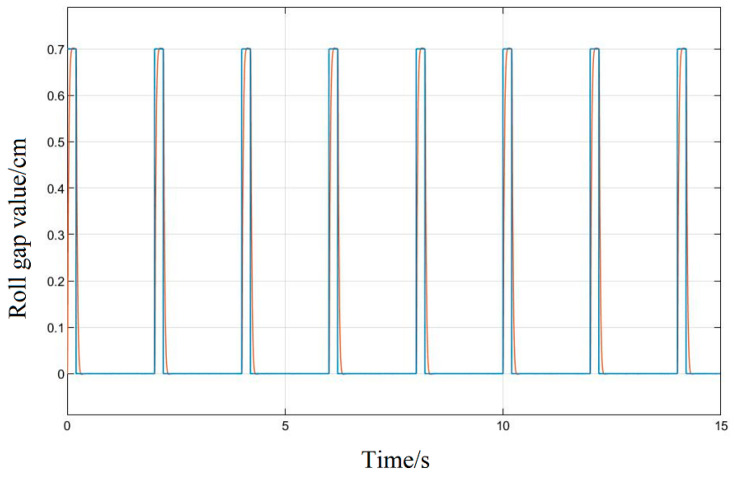
LMPSO method system impulse response diagram.

**Figure 15 biomimetics-08-00143-f015:**
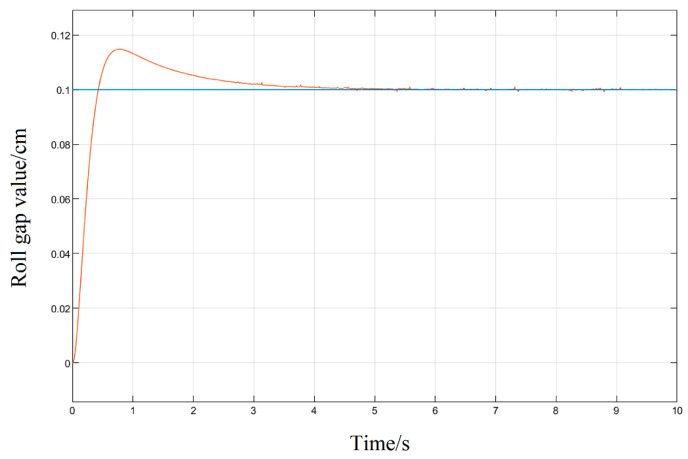
Z-N algorithm control step response diagram.

**Figure 16 biomimetics-08-00143-f016:**
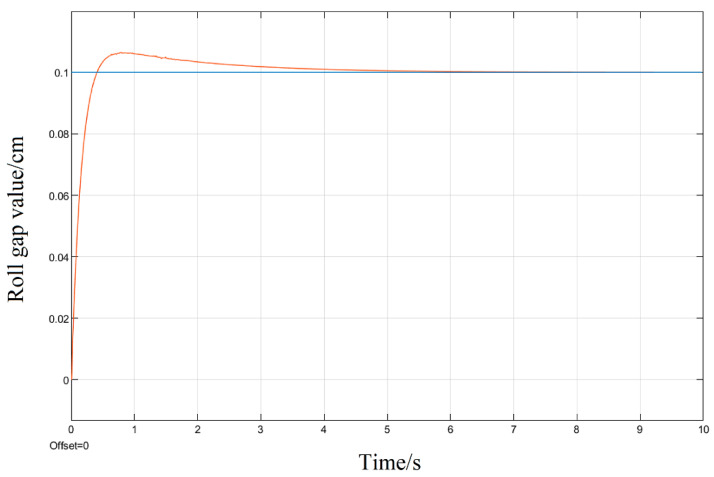
PSO algorithm control step response diagram.

**Figure 17 biomimetics-08-00143-f017:**
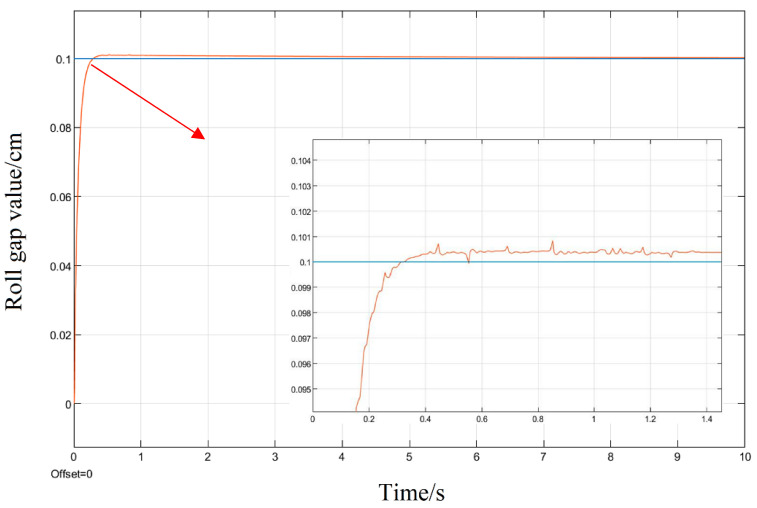
LMPSO algorithm control step response diagram.

**Figure 18 biomimetics-08-00143-f018:**
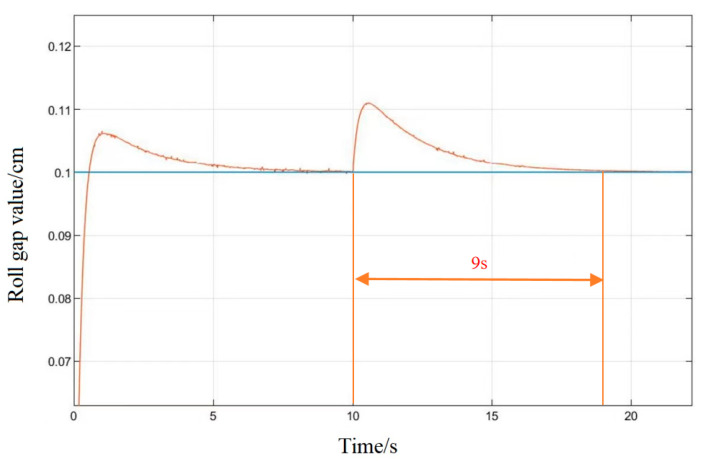
PSO algorithm interference response diagram.

**Figure 19 biomimetics-08-00143-f019:**
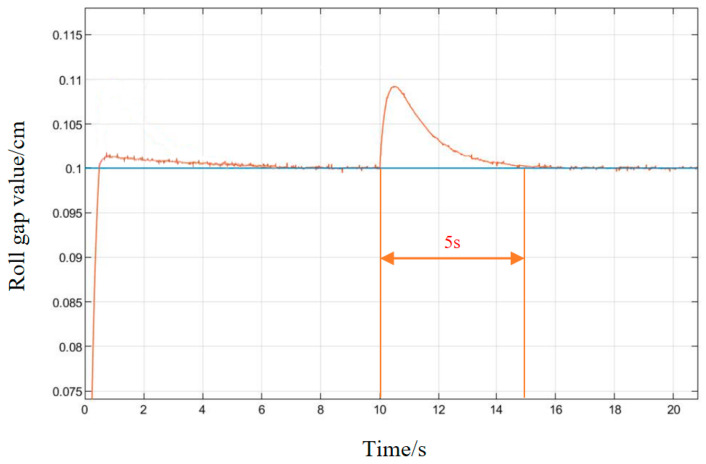
LMPSO algorithm interference response diagram.

**Table 1 biomimetics-08-00143-t001:** Parameters of main components of 5500 mm medium and heavy plate finishing mill.

Parameter	Value
Maximum rolling force	100 MN
Power of main motor	10,000 kW
Mill stand size	15.81 m × 4.57 m × 2.3 m
Servo valve natural frequency	73 Hz
Servo damping ratio $\xi_{v}$	0.543
Mill stand weight	405 ton
Rolling speed	5.36 m/s
Piston effective area	0.177 m^3^
Piston rod diameter	1500 mm
Piston effective stroke	85 mm
Total mass of load piston	4.515 × 105 Kg
Modulus of elasticity of hydraulic cylinder oil	2 × 10^9^ Pa
Displacement sensor gain	1000 $K_{w}/V\cdot m^{-1}$

**Table 2 biomimetics-08-00143-t002:** Performance measures of Z-N, classic PSO and improved PSO.

	Maximum Overshoot (%)	Peak Time (s)	Stable Time (s)
Z-N	15.1	0.73	4.2
PSO	4.92	0.47	4.8
LMPSO	0.84	0.42	0.49

## Data Availability

Not applicable.

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
