# Peer review of "Optimization Strategy of Rolling Mill Hydraulic Roll Gap Control System Based on Improved Particle Swarm PID Algorithm"

_biomimetics, 2023, doi:10.3390/biomimetics8020143_

Round 1

Reviewer 1 Report

In this paper, authors proposed an optimization strategy for rolling mill roll gap PID control. some important points need to be modified.

1.      The study-organizing paragraph is missing. The structure of the introduction section is not good. It should have two separate paragraphs at its end, one of which presents the contribution and explanations of this work.

2.      A summary of the paper content, what are the steps to achieve the final conclusions, is still missing from the introduction section, with a clear emphasis on the novel aspects.

3.      The author states they have not developed the improved part for PSO, I think the improved part is a linear modification weight strategy, it should be developed by other scholar in literature, please confirm it, necessary add the reference.

4.      The manuscript is poorly structured. Please rearrange the structure, especially the title of each part.

5.      There is no result and discussion section in this paper!!!! The authors have finished the manuscript with conclusions!!

6.      Please confirm the copyright of Figure 2.

7.      Please check the writing specifications, for example abbreviation.

Reviewer 2 Report

This paper established a model of a 5500mm thick plate rolling mill thickness automatic control system, optimized the PID parameter setting by improving the particle swarm optimization algorithm, and adopted OPC UA communication technology to realize online semi-physical simulation of Siemens S7-1500 series PLC and MATLAB.

Although the simulation results of the research work show high control accuracy and small overshoot, from my opinion, this manuscript is not acceptable in the present form. The following major observations must be considered before publication:

1)     I strongly suggest to rewrite the Abstract Section. There are some sentences without sense, for example: “The simulation results show that compared with the conventional particle swarm algorithm and the Ziegler-Nichol response curve method, the simulation results of the improved particle swarm algorithm to adjust the PID parameters have high control accuracy and small overshoot.”

2)     The authors mention that under the improved particle swarm algorithm the response speed is fast, the anti-interference ability and robustness are strong, and “followability” (do you mean tracking?) is good. The Reviewer recommend to specify in the Abstract how much faster it is, although some sentences are presented in Section 3.3.

3)     Additionally, the Reviewer suggest to avoid to use terms as robustness, especially when a formal robustness analysis is not provided along the manuscript.

4)     The authors present literature research. In page 1, it is written “Heavy industry has always played an extremely important role in the continuous development of the country. With the rapid progress of science, technology and economy, there are higher requirements for the quality of industrial products.” Maybe, it is better to mention some of the requirements to support the idea of the sentence; this, could emphasizes the novelty of the research work.

5)     Author specifies: “Traditional PID control is difficult to achieve the ideal control effect.” It is highly recommended to add recent references to support this statement. Although PID is well-known, most of the times is enough to control a process, for this reason, authors must specify for which kind of processes it is required advanced control strategies.

6)     If it is possible, please replace “Literature [Reference],” for “In [Reference],”.

7)     At the end of the Introduction Section, when the contribution is showed there are some confusing phrases. For that reason, I suggest to work again on this.

8)     Please use “Simulation Results” instead of “Simulation Research”.

9)     Figure 11 shows the physical map of the simulation platform, please add labels in the figure and do the corresponding reference in pages 10-11, lines 341-356. In addition, a schematic of the connections could be added.

10)  In Section Simulation Research, the Reviewer strongly recommend to use the same axis for Fig. 15, 16 and 17 in order to emphasize the simulation results. For Fig. 17, a zoom image could be added at the top of the figure, if required.

11)  Please write on the correct form the reference of Eqs. 10-14.

12)  In Fig. 7 and Fig. 12-17, increase the quality.

13)  For the Conclusion Section, please work again on this. The Reviewer strongly recommend to present the main results.

Finally, I suggest that a native English person reviews your manuscript, grammatical errors are encountered along the work.

Round 2

Reviewer 1 Report

The authors have addressed all my comments appropriately. 

Reviewer 2 Report

You have done a good job, please change the labels of Eqs. 10-14, they should appear at the right side. Moreover, in Fig. 5 add the negative sign to the feedback scheme. 
